# Zinc/Aluminium–Quinclorac Layered Nanocomposite Modified Multi-Walled Carbon Nanotube Paste Electrode for Electrochemical Determination of Bisphenol A

**DOI:** 10.3390/s19040941

**Published:** 2019-02-22

**Authors:** Rahadian Zainul, Nurashikin Abd Azis, Illyas Md Isa, Norhayati Hashim, Mohamad Syahrizal Ahmad, Mohamad Idris Saidin, Siriboon Mukdasai

**Affiliations:** 1Department of Chemistry, Faculty of Mathematics and Natural Science, Universitas Negeri Padang, West Sumatera 25171, Indonesia; 2Department of Chemistry, Faculty of Science and Mathemathics, Universiti Pendidikan Sultan Idris, Tanjong Malim 35900, Perak, Malaysia; nurashikinazis91@gmail.com (N.A.A.); norhayati.hashim@fsmt.upsi.edu.my (N.H.); syahrizal@fsmt.upsi.edu.my (M.S.A.); idris.saidin@fsmt.upsi.edu.my (M.I.S.); 3Nanotechnology Research Centre, Faculty of Science and Mathemathics, Universiti Pendidikan Sultan Idris, Tanjong Malim 35900, Perak, Malaysia; 4Department of Chemistry, Faculty of Science, Khon Kaen University, Khon Kaen 40002, Thailand; sirimuk@kku.ac.th

**Keywords:** bisphenol A, square-wave voltammetry, layered double hydroxide nanocomposite

## Abstract

This paper presents the application of zinc/aluminium-layered double hydroxide-quinclorac (Zn/Al-LDH-QC) as a modifier of multiwalled carbon nanotubes (MWCNT) paste electrode for the determination of bisphenol A (BPA). The Zn/Al-LDH-QC/MWCNT morphology was examined by a transmission electron microscope and a scanning electron microscope. Electrochemical impedance spectroscopy was utilized to investigate the electrode interfacial properties. The electrochemical responses of the modified electrode towards BPA were thoroughly evaluated by using square-wave voltammetry technique. The electrode demonstrated three linear plots of BPA concentrations from 3.0 × 10^−8^–7.0 × 10^−7^ M (*R*^2^ = 0.9876), 1.0 × 10^−6^–1.0 × 10^−5^ M (*R*^2^ = 0.9836) and 3.0 × 10^−5^–3.0 × 10^−4^ M (*R*^2^ = 0.9827) with a limit of detection of 4.4 × 10^−9^ M. The electrode also demonstrated good reproducibility and stability up to one month. The presence of several metal ions and organic did not affect the electrochemical response of BPA. The electrode is also applicable for BPA determination in baby bottle and mineral water samples with a range of recovery between 98.22% and 101.02%.

## 1. Introduction

Bisphenol A (BPA), is an organic compound consist of two phenol functional groups. It is extensively used as a monomer to manufacture epoxy resins and polycarbonates (PC) plastic [1,2]. There are various ways BPA can be spread into the environment such as via wastewater during the production or via leaching process of epoxy resin and PC plastic products, including food package, water bottles and baby bottles [3]. Further, BPA has been proven by many studies as an endocrine-disrupting chemical [4,5]. By mimicking the role of endogenous hormones, BPA can give an adverse effect on the human central nervous system and reproductive system [6,7]. Besides that, newborns and infants are potentially exposed to BPA because it can also be found in breast milk [8], powdered milk and infant formulas [9]. Until now, the determination of BPA has been carried out by many analytical techniques such as gas chromatography-mass spectrometry [10], spectrophotometric [11], capillary electrophoresis [12], immunoassay detection [13], chemiluminescence detection [14], as well as enzyme-linked immunosorbent assay (ELISA) [15].

Despite the fact that all the analytical techniques named above have high sensitivity with low detection limit, they are relatively expensive instruments and required time-consuming sample pretreatment, making them unsuitable for on-site analysis. Some of them are not user-friendly and need to be by handled by skillful technicians. Since BPA is high electrochemically active, electrochemical sensor acts as an attractive alternative for their determination, possess the merit of simple operating, fast response, economical, and suitable for on-site analysis. However, the electron transfer of BPA at bare electrodes is slow, which further can affect their sensitivity. For that reason, some functional materials have been introduced to develop BPA sensor with high sensitivity of detection such as Au nanoparticles [16,17,18], NiO nanocomposite [19,20], ferroferric oxide nanoparticles [21,22] and Bi_2_WO_6_ nanoplate [23] but the sensor still provided a narrow working range. The layered material nanocomposites with the magnetic properties of the metal are remarkably sensitive to the intercalation of molecular anions. For example, the presence of zinc in nanocomposite aided in the intercalation of hydroxide-2(3-chlorophenoxy) propionate and 2(3-chlorophenoxy) propionate resulting in the material which feasible for sensor applications [24,25,26]. Recently, the intercalation of quinclorac (QC) with host material layered double hydroxide (LDH) has potential to be used as a modifier for electrode [27]. QC, which is known as 3,7-dichloro-8-quinolinecarboxylic acid is a type of herbicide that was broadly used in paddy cultivation. Besides that, it is beneficial in controlling of some broad leaf weeds and grass in crops [28]. The chemical structure of QC is shown in Figure 1.

A number of carbon-based materials such as graphite, glassy carbon (GC), carbon fibers and carbon nanotubes (CNTs) are commonly used in electrochemical analysis as conductive materials. These materials are extensively employed in sensor fabrication because they have a wide potential window, rich surface chemistry, chemical inertness, and also economical [29]. Among them, carbon nanotubes (CNTs) have gained much attention in the preparation of a mixture of carbon paste electrode (CPE). One of the CNTs types which are multiwalled carbon nanotubes (MWCNTs), has many advantages such as small in size, high specific surface area, high chemical stability, high electrical and thermal conductivity and high mechanical strength, making them superiority over other types of carbon-based materials [30]. High surface area to volume ratio materials are more exposed for reaction and they will react at a much faster rate, the further effect on improved mass transport characteristics. This will promote the electron transfer between the electroactive species and the electrode surface [31]. Compared to other small metallic structures, the MWCNTs are higher in term of stability against electromigration [32].

There is a need for developing an economical and simple yet high-sensitivity method for determination of BPA. For that purpose, this paper presents the modification of MWCNT with layered material nanocomposite of zinc/aluminium–quinclorac for determination of BPA by using square-wave voltammetry technique.

## 2. Materials and Methods

### 2.1. Chemicals and Reagents

BPA used was from Sigma-Aldrich Co. (St. Louis, MO, USA). A stock solution of 0.01 M BPA was prepared with ethanol (95% v/v) and kept in a refrigerator at 4 °C. Working solutions were freshly prepared before use by diluting the stock solution in supporting electrolyte. Potassium phosphate buffer solution (PBS) was prepared by 0.1 M KH_2_PO_4_ and 0.1 M K_2_HPO_4_. MWCNT (Timesnano, Chengdu, China) and all analytical grade reagents were used as received without further purification. All the solutions were prepared with deionized water (EASY pure LF, Barnstead, Essex, UK). 

### 2.2. Apparatus

Square-wave voltammetry (SWV) and electrochemical impedance spectroscopy (EIS) studies were conducted using a potentiostat, model Series-G750 and model Ref 3000, respectively. Zn/Al-LDH-QC/MWCNT paste electrode acted as a working electrode, while both reference and counter electrodes were represented by platinum wire and Ag/AgCl, respectively. The morphological studies of the Zn/Al-LDH-QC/MWCNT were examined by scanning electron microscope (SEM) and transmission electron microscope (TEM), model SU8020 UHR (Hitachi, Japan). The pH measurements were carried out using Thermo Scientific Orion 2-Star Benchtop pH Meter (Thermo Fisher Scientific, Chelmford, MA, USA), which was calibrated with standard pH buffer solutions.

### 2.3. Synthesis of Zn/Al-LDH-QC Nanocomposite

By using Zn(NO_3_)_2_ and Al(NO_3_)_3_ as precursors, the Zn/Al-LDH were synthesized by a conventional co-precipitation method as previously reported [27]. Both precursors were dissolved in 250 mL of deionized water. 50 mL (0.05 M, 0.1 M and 0.2 M) of QC solution was then added and the mixture was stirred. The pH of the solution was adjusted to 7.5 with sodium hydroxide. The slurry was aged for 24 h in oil bath shaker at 70 °C. The resulting precipitate was then centrifuged, washed by deionized water and dried in an oven at 60 °C. Finally, the dried precipitate of Zn/Al-LDH-QC was finely ground and kept in a glass bottle at room temperature until the subsequent experiment.

### 2.4. Preparation of Electrodes

A paste of the proposed electrodes was prepared by mixing Zn/Al-LDH-QC (0%, 5%, 10%, 15%) and MWCNT (100%, 95%, 90%, 85%) via hand mixing in a mortar and pestle (Figure 2). The mixture was then added with two drops of liquid binder (paraffin oil). Next, the homogenized paste obtained was firmly packed into a Teflon tube (i.d. 2.0 mm). One end of the tube was connected to copper wire in order to produce an electrical contact between paste and wire. Prior to each measurement, the electrode surface was smoothened with soft paper. 

### 2.5. Measurement Procedure

Voltammetric measurements were performed in the desired concentration of BPA solution containing 0.1 M PBS, at pH 7.0. Before each measurement, the solutions were deoxygenated by nitrogen gas purge for 15 minutes. The voltammetric measurement was recorded using SWV in the potential range from −0.2 V to +1.0 V at 150 Hz frequency, 6 mV step increment, and 60 mV pulse height. All the experiments were carried out at room temperature of 25 ± 2 °C.

### 2.6. Real Samples

A baby bottle was thoroughly washed prior the analysis. The bottle was filled with deionized water and heated for 1 h at 70 °C and 4 hours at 30 °C. Then, the baby bottle was left for a week at room temperature. The solution was then added with 1.0 mL of ethanol. Finally, the solution was filtrated and diluted with PBS. Two mineral water samples were purchased. The first mineral water sample was left for two weeks in a car exposed to sunlight, while the other mineral water sample was filled with deionized water and heated for 5 h at 60 °C. Then both samples were directly tested without any pretreatments.

## 3. Results

### 3.1. Surface Morphology Studies of Zn/Al-LDH-QC-MWCNT

SEM and TEM were used to examine the morphology of Zn/Al-LDH-QC and Zn/Al-LDH-QC/MWCNT. From the previous report on characterization of Zn/Al-LDH and Zn/Al-LDH-QC, Figure 3A indicates a smaller and irregular particle size of Zn/Al-LDH-QC with a fine plate-like characteristics [27]. The transparent delicate tube in Figure 3B representing MWCNT and they were covered by opaque features of Zn/Al-LDH-QC.

### 3.2. Electrode Characterization

Figure 4 shows a well-defined cyclic voltammogram (CV) in 4.0 mM K_3_[Fe(CN)_6_]. Compared to the MWCNT paste electrode (curve a), the Zn/Al-LDH-QC/MWCNT paste electrode (curve b) showed increasing in redox currents. Both anodic peak current (I_pa_) and cathodic peak current (I_pc_) calculated from MWCNT paste electrode were 5.064 μA and 2.873 μA, respectively. Meanwhile, the redox peak current of Zn/Al-LDH-QC/MWCNT paste electrode was enhanced to I_pa_ = 7.962 μA and I_pc_ = 4.040 μA. Additionally, peak-to-peak separation (ΔE_p_) of Zn/Al-LDH-QC/MWCNT paste electrode also decreased from 0.366 V to 0.333 V. These results suggested that the implementation of modifier Zn/Al-LDH-QC in the MWCNT paste electrode contributed to improving the electrochemical response of the electrode in terms of electron-transfer rate and conductivity. 

Electrochemical impedance spectroscopy (EIS) was performed to evaluate the capability of electron transfer of the electrode. Generally, Nyquist plot of the EIS consists of two portions which are a semicircle portion and a linear portion. The linear portion corresponds to the diffusion process. The semicircle diameter represents the electron-transfer resistance (*R*_ct_) at the surface of electrode [33]. Figure 5 illustrated the Nyquist plot of MWCNT paste electrode (curve a) and Zn/Al-LDH-QC/MWCNT (curve b) paste electrode. Both electrodes showed almost straight lines, revealing that Zn/Al-LDH-QC attributed the ability good electron transfer. By fitting the Randles equivalent electrical circuits (Figure 5 inset), the *R*_ct_ values for the unmodified MWCNT paste electrode was 34.03 ohms. After the modification of the electrode using Zn/Al-LDH-QC, the value of *R*_ct_ decreased to 0.101 ohms, which was smaller than the unmodified MWCNT paste electrode. It was contributed by the presence of high conductive modifier in the MWCNT paste electrode.

In addition, the electron transfers apparent rate constant (*k*_apps_) value was calculated from Equation (1);
(1)$$k_{apps}=\frac{RT}{F^{2}R_{\text{ct}}C}$$ where, *R* = Gas constant (8.314 J mol^−1^ K^−1^); *T* = Absolute temperature (298 K); F = Faraday constant (96485 C mol^−1^); *C* = K_3_[Fe(CN)]_6_ solution concentration.

The *k*_apps_ values calculated for MWCNT paste electrode and Zn/Al-LDH-QC/MWCNT paste electrode were 1.96 × 10^−6^ cm s^−1^ and 6.65 × 10^−4^ cm s^−1^, respectively. The Zn/Al-LDH-QC/MWCNT paste electrode showed high *k*_apps_ and low *R*_ct_ values which indicates a faster electron-transfer process. This process has been speeding up by the Zn/Al-LDH-QC. The EIS result consents very well with the CV findings. Scheme 1 illustrates the electro-oxidation process of BPA which involved two electrons and two protons, and the oxidation reaction of BPA on the electrode surface during electrochemical sensing.

### 3.3. Optimization of the Experimental Conditions

#### 3.3.1. Effect of Modifier Content

In this experiment, the SWV was employed to study the electrochemical performance of different composition ratios (% w/w) between Zn/Al-LDH-QC and MWCNT in 0.1 mM BPA. As illustrated in Figure 6, the current response increased with composition ratios of Zn/Al-LDH-QC in MWCNT paste electrode and achieved the maximum at 10%, further increase the percent of modifier caused a current response to decrease. This phenomenon corresponds to an excessive amount of Zn/Al-LDH-QC in MWCNT paste electrode would reduce the conductivity of the surface of the electrode and their charge transfer kinetics. Besides that, the excessive amount of modifier will form a thick film and further limited the mass transport of BPA at electrode surface [34]. Therefore, the modified MWCNT paste electrode with 10% of Zn/Al-LDH-QC was implemented for subsequent experiments. 

#### 3.3.2. Effect of pH

The electrochemical response may be affected by pH of the analyte solution. In this study, the current peak of BPA increased from pH 6.0 to 7.4 (Figure 7). Meanwhile, at pH exceed 7.4, the current peak started to decline. This could be ascribed to the competition that occurred between hydroxyl anion and BPA molecule on the surface of the electrode. Therefore, pH 7.4 was used in all the following experiments. Besides that, increasing of pH also caused negatively shifting of the peak potential. The equation of this relationship can be derived as of *E*_pa_ (V) = −0.0541 pH + 0.9139 (*R*^2^ = 0.9942). The 0.0541 V/pH shift obtained was approximately near to the Nernst value (0.0576 V/pH) indicating a balanced number of protons and electrons transfer in the electrode reaction.

#### 3.3.3. Effect of Electrolytes

The effect of several types of supporting electrolytes such as NaCl, KCl, KNO_3_, Na_2_SO_4_, CH_3_COONa, and PBS towards BPA response was studied. As shown in Figure 8, the SWV voltammogram displays a higher *I*_pa_ and better peak shape in PBS compared to the other supporting electrolytes, thus PBS was chosen as suitable supporting electrolyte for further studies. 

#### 3.3.4. Effect of SWV Parameters

Based on the calculations performed by Osteryoung, SWV characteristics are influenced by some parameters that applied in the experiments [35]. Therefore, SWV parameters such as frequency, pulse size, and step size were further investigated in detail. The influence of frequency (30 to 160 Hz) towards current response was investigated (Figure 9A), and the peak current was found to be linearly to frequency. Therefore, 160 Hz was chosen as an optimum frequency as it generated a very good resolution. As shown in Figure 9B, the range of pulse size between 20 mV and 70 mV was evaluated, and the highest current peak exhibited by pulse size 60 mV. The increasing of step size from 1 mV to 6 mV caused the current peak to increase (Figure 9C), hence, 6 mV was chosen as the optimum step size with a sharper peak. Therefore, the optimum SWV conditions (frequency: 160 Hz, pulse size: 60 mV, step size: 6 mV) were applied for better sensitivity and resolution. 

### 3.4. Effects of Scan Rate

Figure 10 shows a cyclic voltammogram of 0.1 mM BPA at Zn/Al-LDH-QC/MWCNT paste electrode with different scan rate (25–250 mV s^−1^). The current peak increased successively when the scan rate increase. Meanwhile, the inset graph shows the linear rising of the current peak with the increase of scan rate. The relationship can be described as: *I*_pa_ (μA) = 0.0778 *E* (mV s^−1^) + 3.802 (*R*^2^ = 0.9825), which classified the BPA oxidation on Zn/Al-LDH-QC/MWCNT as an adsorption-controlled process. 

### 3.5. Chronocoulometry Studies

Chronocoulometry studies were performed in 4.0 mM K_3_[Fe(CN)_6_] to investigate the effective electrochemical surface area of the MWCNT paste and Zn/Al-LDH-QC/MWCNT paste electrodes, based on the Anson equation (Equation (2));
(2)$$Q(t)=\frac{2nFAcD^{\frac{1}{2}}t^{\frac{1}{2}}}{\pi^{\frac{1}{2}}}+Q_{\text{dl}}+Q_{\text{ads}}$$ where, *Q*(*t*) = Charge (Coulombs); *n* = Number of electron transfer; F = Faraday constant (96485 Coulombs/mole); *A* = Effective electrochemical surface area (cm^2^); *c* = Concentration of substrate (mole/cm^3^); *D* = Diffusion coefficient of K_3_[Fe(CN)_6_] ( 7.6 × 10^−6^ cm^2^ s^−1^); *t* = Time (s); *Q*_dl_ = Double layer charge (Coulombs); *Q*_ads_ = Faradaic charge (Coulombs).

According to the slopes of the plots of *Q* vs. *t*^1/2^ (Figure 11A), the effective electrochemical surface area for MWCNT paste electrode was calculated as 1.103 cm^2^, and 1.660 cm^2^ for Zn/Al-LDH-QC/MWCNT paste electrode. These indicated that the modification of MWCNT paste electrode with Zn/Al-LDH-QC increased the electrode effective surface area. This phenomenon would increase the site of BPA oxidation, increase current response, and further improve the electrochemical performance of the sensor [36]. Then, the chronocoulometry experiments were carried out on Zn/Al-LDH-QC/MWCNT paste electrode in 0.1 mM BPA. After background subtraction, the plot of charge (*Q*) against the square root of time (*t*^1/2^) (Figure 11B), exhibited a linear relationship with slope and *Q*_ads_ of 1.98 × 10^−5^ C and 1.47 × 10^−5^ C, respectively. Hence, *D* was calculated to be 5.47 × 10^−4^ cm^2^ s^−1^. Based on the Cottrell equation, *Q*_ads_ = *nFAΓ*_s_, the adsorption capacity, *Γ*_s_, can be obtained as 4.58 × 10^−11^ mol cm^−2^. The *Γ_s_* was higher than LDH/GCE (1.10 × 10^−11^ mol cm^−2^) [37] and PAMAM/Fe_3_O_4_/GCE (4.22 × 10^−11^ mol cm^−2^) [38]. This contributes to a good detection of BPA due to high surface coverage owned by the modified electrode. 

### 3.6. Calibration Curve

The square-wave voltammogram in Figure 12 shows that *I*_pa_ increases linearly with concentration of BPA. The peak currents were linear with the concentration of BPA over three intervals in the range of 3.0 × 10^−8^ – 7.0 × 10^−7^ M (*R*^2^ = 0.9876), 1.0 × 10^−6^–1.0 × 10^−5^ M (*R*^2^ = 0.9836), and 3.0 × 10^−5^–3.0 × 10^−4^ M (*R*^2^ = 0.9827). The equation expressed as *I*_pa_ (μA) = −0.3801 log [BPA] + 3.206, *I*_pa_ (μA) = −2.076 log [BPA] + 13.26, and *I*_pa_ (μA) = −12.66 log [BPA] + 61.95. The detection limit was 4.4 × 10^−9^ M (S/N = 3). The data obtained were compared with the other related published papers as listed in Table 1. Scheme 2 illustrates the reaction mechanism occurred at the Zn/Al-LDH-QC/MWCNT paste electrode and BPA solution. In a solution, BPA was oxidized by releasing two electrons and protons, while in an electrode surface, QC in an interlayer Zn/Al-LDH accepting those electrons and protons.

### 3.7. Reproducibility, Stability and Interferences

There are many parameters influenced the sensor performance, mainly the reproducibility and the stability of the electrode. To evaluate the reproducibility of Zn/Al-LDH-QC/MWCNT paste electrodes, triplicate Zn/Al-LDH-QC/MWCNT paste electrodes were prepared and applied in the determination of in 0.1 mM BPA. The relative standard deviation (RSD) obtained was 1.92% (*n* = 3), indicates a high reproducible of sensor preparation method. Moreover, after one month, the current response of 0.1 mM BPA retained 96.6% of the initial current indicates that the modified has good stability. The selectivity of the modified electrode was investigated by measuring the response 0.1 mM BPA in the existence of some potential coexisting substances such as Al^3+^, Cu^2+^, K^+^, Mg^2+^, Na^+^, Ni^2+^, Zn^2+^, NO_3_^−^, SO_4_^2−^, dopamine, acetaminophen, and ethanol. Figure 13 summarizes the effect of 1.0 mM and 5.0 mM excess concentration of the possible interferences towards the detection of 0.1 mM BPA. Overall, the peak current change was less than ± 15%, which ascribed a good anti-interference ability of the modified electrode. 

### 3.8. Real Samples

The feasibility of Zn/Al-LDH-QC/MWCNT paste electrode in the determination of BPA in real samples was evaluated by performing the analysis in baby bottle and mineral water samples. Since BPA is not detected from both samples, the sample solutions were spiked with standard BPA solutions at different concentrations. Percent recoveries found were in the range of 98.22% and 101.02% (Table 2). These indicate good reliability of the sensor for real samples analysis.

## 4. Conclusions

The role of Zn/Al-LDH-QC as a mediator in BPA determination was successfully demonstrated. The proposed electrode shows a good conductivity with high electron-transfer rate through CV and EIS studies. The effective electrochemical surface area of the electrodes was also investigated by chronocoulometry studies. The electrode exhibits a large effective electrochemical surface area and high adsorption capacity. At optimal condition, the electrode shows a wide linear working concentration range with detection limit of 4.4 nM. The electrode is also applicable for BPA analysis in baby bottle and mineral water samples with good recoveries.

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
