# Peer review of "Zinc/Aluminium–Quinclorac Layered Nanocomposite Modified Multi-Walled Carbon Nanotube Paste Electrode for Electrochemical Determination of Bisphenol A"

_sensors, 2019, doi:10.3390/s19040941_

Round 1
Reviewer 1 Report
The results and discussion presented are good but the subject already extensively studied. Then is not any novelty in electrochemical proposed comparing anothers papers published.
The impedance results should be better discussed. The fit of the curve should be displayed. It should be explained why, for example, the resistance of the solution is different in curves a and b of Figure 4.
The influence of others phenol molecules interference should be studied, such as, cathecol, hydroquinone, phenol
The low LOD is for the lowest concentration range, but in this range the sensitivity is very low.
Discuss the three ranges of linearity of the analytical curve. What is happening on the surface to explain this behavior.
Author Response
Response to reviewers
Thank you for your precious time to review our manuscript. We sincerely appreciate the very constructive comments and suggestions offered. They provide us with a new perspective on improving the manuscript and its impact. All questions have been addressed to in the manuscript.
We also revised the whole manuscript carefully and tried to avoid any grammar or syntax error. In addition, we have asked several colleagues who are skilled authors of English language papers to check the English.
Reviewer 1:
Q1:
The results and discussion presented are good but the subject already extensively studied. Then is not any novelty in electrochemical proposed comparing another papers published.
A1:
Thank you for a comment. Although the determination of bisphenol A already extensively studied, but the application of layered double hydorxide quinclorac in the modification of carbon paste electrode have not been published yet. Thus, the novelty of this studies is the application of zinc/aluminium-layered double hydroxide-quinclorac (Zn/Al-LDH-QC) as a modifier of multiwalled carbon nanotubes (MWCNT) paste electrode for the determination of bisphenol A (BPA).
Q2:
The impedance results should be better discussed. The fit of the curve should be displayed. It should be explained why, for example, the resistance of the solution is different in curves a and b of Figure 4.
A2:
Thank you for the suggestion. We have already displayed the fit of the curve (Figure 5) and extensively discussed the impedance results such as the differences of resistance occur between the two types of electrode rather than the solution in line 177-182.
Q3:
The influence of others phenol molecules interference should be studied, such as, cathecol, hydroquinone, phenol.
A3:
Thank you for the suggestion. We have already added the influence of other phenol compounds in the interference study. However, since this study performed the real sample analysis in mineral water and baby bottle to evaluate the feasibility of the modified electrode, we are considered the influence of different types of metal (line 416) rather than the others phenol molecules.
Q4:
The low LOD is for the lowest concentration range, but in this range the sensitivity is very low.
A4:
Thank you for a comment. We have already corrected the sentences and the formula used to calculate LOD.
Q5:
Discuss the three ranges of linearity of the analytical curve. What is happening on the surface to explain this behavior.
A5:
Thank you for the suggestion. We have already clarified the three ranges of linearity as the three intervals (line 362). The reaction occured on the surface of the electrode have already explained in line 367 and illustrated in Scheme 2.
Reviewer 2 Report
The manuscript presents a new sensor for the determination of Bisphenol A using square wave voltammetry. The discussed electrode is based on Zn/Al-Quinclorac nanocomposite modified with carbon nanotubes (CNTs).
The introduction presents the necessity of bisphenol A determination in various samples listing the available analytical methods and sensors used for that purpose. It also lists the advantages of using CNTs in sensors applications. However:
- the introduction lacks the information about the quinclorac (chemical formula, applications, etc.).
- in line 50 authors state that "all analytical techniques named above ... are relatively expensive and require time-consuming sample pretreatment ...". This does not apply to electrochemical sensors mentioned in line 49.
- The presented work is a continuation of earlier research on Zn/Al-Quinclorac nanocomposite (ref. 32) and it should be clearly stated already in the introduction.
The "Materials and methods" part is written clearly including all the information necessary for repeating the experiment. Two minor comments:
- in line 106, please provide the information about how long the nanocomposite was stored and the material of a container.
- editing error in lines 108/109, value (15%) is missing.
The results part must be improved by addressing following issues:
- In part 3.1., few sentences about the comparison of morphology of Zn/Al-QC and Zn/Al pastes should be included (note that some readers might not be familiar with your previous work in ref. 32)
- Information about the reaction responsible for voltammetric peaks should be described in the text and schemes 1 and 2 (preferably combined as one scheme) should be included already in part 3.2., rather than later in the text.
- In the Nyquist plot (fig. 4.) the axes represent real and imaginary parts of impedance and should be denoted as Z' and Z'' (not Z and Z')
- In lines 242-243, the theoretical value is mentioned. Where was it obtained? The reference should be provided
- (Most important issue) In the discussion of SWV parameters namely frequency and step size, the optimal values seems to be the maximal values of the chosen experimental range. More experiments should be performed or the choice of particular parameters range should be explained.
- In Anson equation there should be "Q(t) = " instead of "(t)"
- In part 3.7., the molar concentrations of the used interferents should be provided.
Author Response
Response to reviewers
Thank you for your precious time to review our manuscript. We sincerely appreciate the very constructive comments and suggestions offered. They provide us with a new perspective on improving the manuscript and its impact. All questions have been addressed to in the manuscript.
We also revised the whole manuscript carefully and tried to avoid any grammar or syntax error. In addition, we have asked several colleagues who are skilled authors of English language papers to check the English.
Reviewer 2:
Q1:
The manuscript presents a new sensor for the determination of Bisphenol A using square wave voltammetry. The discussed electrode is based on Zn/Al-Quinclorac nanocomposite modified with carbon nanotubes (CNTs).
A1:
Thank you for the comments.
Q2:
The introduction presents the necessity of bisphenol A determination in various samples listing the available analytical methods and sensors used for that purpose. It also lists the advantages of using CNTs in sensors applications. However:
a) the introduction lacks the information about the quinclorac (chemical formula, applications, etc.).
A2 (a):
Thank you for the suggestion. We have already added more information about the quinclorac in the introduction section (line 63), as well as its chemical structure (Figure 1).
b) in line 50 authors state that "all analytical techniques named above ... are relatively expensive and require time-consuming sample pretreatment ...". This does not apply to electrochemical sensors mentioned in line 49.
A2 (b):
We are apologise for the mistake. We have already corrected the subject mentioned by removing ‘electrochemical sensors’ from line 48.
c) The presented work is a continuation of earlier research on Zn/Al-Quinclorac nanocomposite (ref. 32) and it should be clearly stated already in the introduction.
A2 (c):
We agree and appreciate the suggestion. We have already highlighted the continuation of this work with the previous research on Zn/Al-Quinclorac nanocomposite in the introduction by citation (line 63).
Q3:
The "Materials and methods" part is written clearly including all the information necessary for repeating the experiment. Two minor comments:
a) in line 106, please provide the information about how long the nanocomposite was stored and the material of a container.
A3 (a):
Thank you for the suggestion. We already provided the informations in line 113.
b) editing error in lines 108/109, value (15%) is missing.
A3 (b):
We are apologise for the editing error. We have already added the missing subject in line 117.
Q4:
The results part must be improved by addressing following issues:
a) In part 3.1., few sentences about the comparison of morphology of Zn/Al-QC and Zn/Al pastes should be included (note that some readers might not be familiar with your previous work in ref. 32)
A4 (a):
Thank you for the suggestion. We already added a sentences about the morphology of Zn/Al-LDH-QC from previous work in line 140.
b) Information about the reaction responsible for voltammetric peaks should be described in the text and schemes 1 and 2 (preferably combined as one scheme) should be included already in part 3.2., rather than later in the text.
A4 (b):
Thank you for the suggestion. We have already made an amendment as suggested (line 211), and combined both Schemes 1 and 2.
c) In the Nyquist plot (fig. 4.) the axes represent real and imaginary parts of impedance and should be denoted as Z' and Z'' (not Z and Z')
A4 (c):
We are apologise for the denoting error. We have already corrected Z and Z’ in the Nyquist plot (fig. 4.) as Z’ and Z”.
d) In lines 242-243, the theoretical value is mentioned. Where was it obtained? The reference should be provided
A4 (d):
Thank you for a comment. We have already provided the name of the theoretical value in line 261.
e) (Most important issue) In the discussion of SWV parameters namely frequency and step size, the optimal values seems to be the maximal values of the chosen experimental range. More experiments should be performed or the choice of particular parameters range should be explained.
A4 (e):
Thank you for the comments. We have already explained the choice of particular parameters namely frequency and step size in line 304 and 307, respectively.
f) In Anson equation there should be "Q(t) = " instead of "(t)"
A4 (f):
We are apologise for the error. We have already corrected “(t)” in the Anson equation as “Q(t)” in Equation 2.
g) In part 3.7., the molar concentrations of the used interferents should be provided.
A4 (g):
Thank you for a comment. We have already provide the molar concentration of the used interferents in the manuscript (line 418).